# PixNerd: Pixel Neural Field Diffusion

**Shuai Wang**[1]    **Ziteng Gao**[3]    **Chenhui Zhu**[1]    **Weilin Huang**[2]    **Limin Wang**[1,✉]

[1]Nanjing University    [2]ByteDance Seed    [3]National University of Singapore

`https://github.com/MCG-NJU/PixNerd`
`https://huggingface.co/spaces/MCG-NJU/PixNerd`

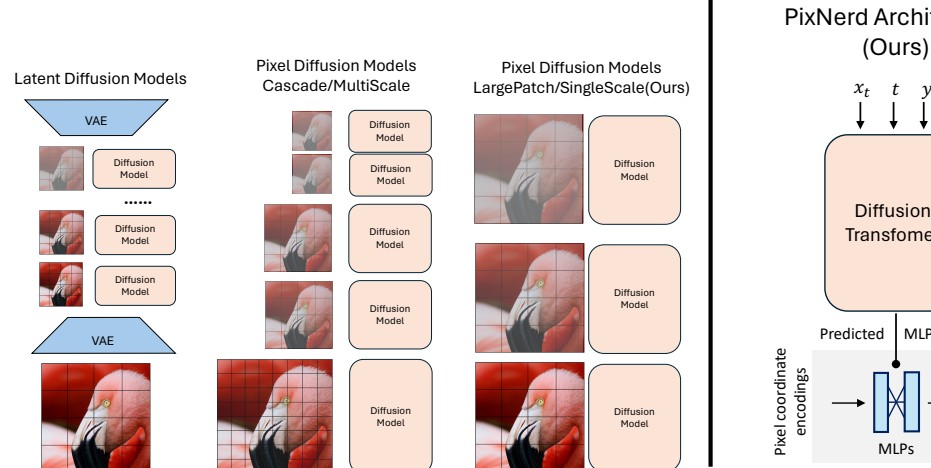

Figure 1: **Left: Comparison with other diffusion paradigms. Right: PixNerd architecture.**
Our PixNerd is a streamlined LargePatch/SingleScale pixel space diffusion model. PixNerd follows the diffusion transformer design, replacing the final linear projection with a neural field to model the large patch details.

## Abstract

The current success of diffusion transformers is built on the compressed latent space shaped by the pre-trained variational autoencoder (VAE). However, this two-stage training paradigm inevitably introduces accumulated errors and decoding artifacts. To address these issues, researchers have returned to pixel space modeling but at the cost of complicated cascade pipelines and increased token complexity. Motivated by the simple yet effective diffusion transformer architectures on the latent space, we propose to model pixel space diffusion using a large-patch diffusion transformer and employ neural fields to decode these large patches, resulting in a single-stage streamlined end-to-end solution. We coin our method as pixel neural field diffusion transformer (**PixNerd**). Thanks to the efficient neural field representation in PixNerd, we achieve **1.93 FID** on ImageNet $256 \times 256$ and **nearly** $8\times$ **lower latency** without any complex cascade design or VAE compared with these previous pixel diffusion models. We also extend our PixNerd framework to text-to-image tasks, and achieve a competitive 0.73 overall score on the GenEval benchmark and 80.9 overall score on the DPG benchmark.

## 1 Introduction

The current success of diffusion transformers largely depends on variational autoencoders (VAEs) (Rombach et al., 2022; Yao & Wang, 2025; Chen et al., 2024a). VAEs significantly reduce the spatial dimension of raw pixels while providing a compact and nearly lossless latent space, substantially easing the learning difficulty for diffusion transformers. Operating in this compressed space, diffusion transformers can effectively learn the reverse diffusion process using small patch sizes. However, training high-quality VAEs typically requires adversarial training (Brock et al.,

---
✉ Corresponding author: Limin Wang (lmwang@nju.edu.cn).

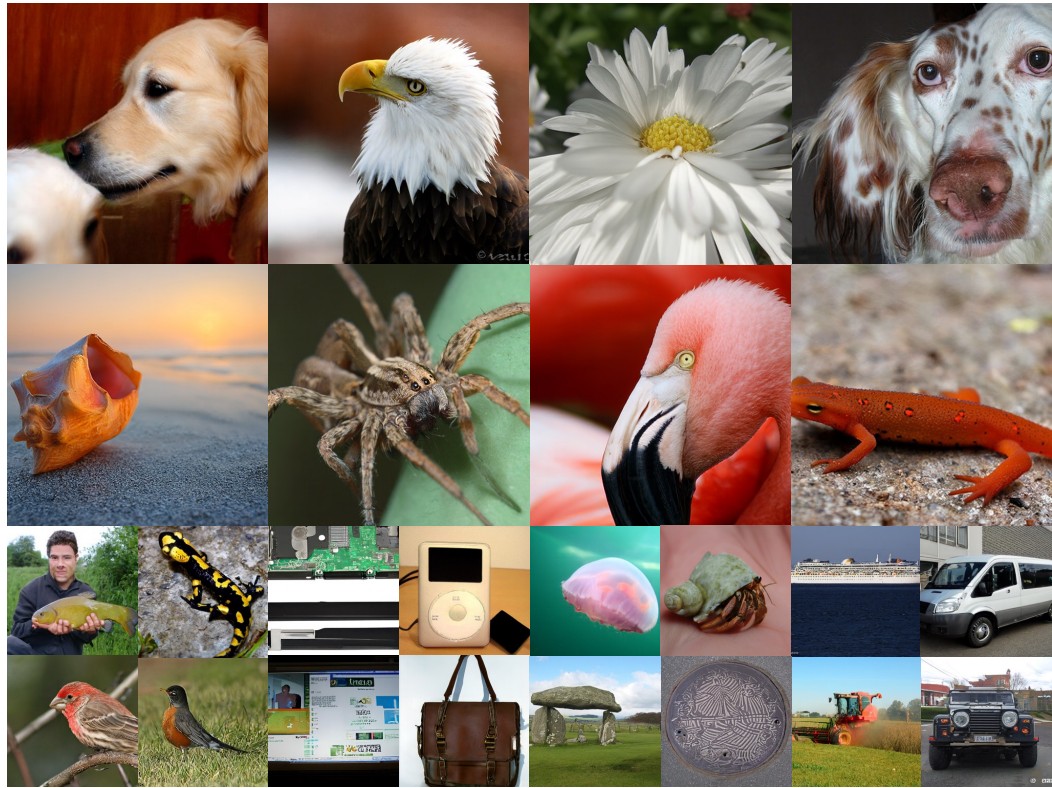

Figure 2: **Selected** $256 \times 256$ **and** $512 \times 512$ **resolution samples.** Generated from PixNerd-XL/16 trained on ImageNet $256 \times 256$ resolution and ImageNet $512 \times 512$ resolution with CFG = 3.5.

2018; Sauer et al., 2022; Rombach et al., 2022; Yao & Wang, 2025; Chen et al., 2024a) and additional supervision (Zhang et al., 2018), introducing complex optimization challenges. Moreover, this two-stage training paradigm leads to accumulated errors and inevitable decoding artifacts.

To address these limitations, Leng et al. (2025) has explored joint training approaches, though these come with substantial computational costs. An alternative approach involves implementing diffusion models directly in raw pixel space (Dhariwal & Nichol, 2021; Chen et al., 2025b; Teng et al., 2023). In contrast to the success of latent diffusion transformers, progress in pixel-space diffusion transformers proves significantly more challenging. Without the dimensionality reduction provided by VAEs, pixel diffusion transformers allocate substantially more image tokens (Chen et al., 2025b), requiring impractical computational resources when using the same patch size as latent diffusion transformers. To maintain a comparable number of image tokens, pixel diffusion transformers must employ much larger patch sizes. Unfortunately, due to the vastness of raw pixels, larger patches make diffusion learning particularly difficult. Chen et al. (2025b); Teng et al. (2023) have proposed cascade solutions that divide the diffusion process across different scales to reduce computational costs. However, such a cascade approach complicates both training and inference.

In contrast to the afore-mentioned methods, we explore the performance upper bound using a large-patch diffusion transformer on raw pixels while **maintaining the same token count and comparable computational requirements** as latent diffusion models. Inspired by the success of implicit neural fields in scene reconstruction (Mildenhall et al., 2021; Sitzmann et al., 2020), we propose modeling large-patch decoding using an implicit neural field in the pixel diffusion pipeline. We replace the traditional diffusion transformer's linear projection with an patch-wise implicit neural field head, leading to a novel pixel-space architecture **PixNerd** (**Pix**el **Ner**ual Field **D**iffusion).

Specifically, we faithfully follow the classic diffusion transformer and design a patch-wise adaptive neural field head, whose weights are predicted by the diffusion transformer's last hidden features. For each pixel within a patch, we first transform its local coordinates along with noisy pixel values into input encoding, and then predict the diffusion velocity by the neural field MLP. Our approach significantly alleviates the challenge of learning fine details with large-patch configurations.

Compared to previous latent diffusion transformers and pixel-space diffusion models, our end-to-end PixNerd offers a more streamlined and efficient solution for image generative tasks. For class-conditional image generation on ImageNet $256 \times 256$, PixNerd-XL/16 achieves a competitive 1.93 FID and significantly better 4.50 sFID (indicating superior spatial structure) than others. For text-to-image generation, PixNerd-XXL/16 achieves 73.0 on the GenEval benchmark and 80.9 average score on the DPG benchmark.

Our contributions are summarized as follows:

- We propose a new, efficient end-to-end pixel-space diffusion transformer equipped with our proposed neural field head, dubbed PixNerd.
- For the class-to-image generation, on ImageNet $256 \times 256$, our PixNerd-XL/16 achieves a competitive 1.93 FID with similar computation demands as its latent counterpart.
- For the text-to-image generation, our PixNerd-XXL/16 achieves a 0.73 overall score on the GenEval benchmark and 80.9 average score on DPG benchmark.

## 2 RELATED WORK

**Latent Diffusion Models**   train diffusion models on a compact latent space shaped by a VAE. Compared to raw pixel space, this latent space significantly reduces spatial dimensions, easing both learning difficulty and computational demands (Rombach et al., 2022; Yao & Wang, 2025; Chen et al., 2024a). Thus VAE has become a core component in modern diffusion models (Peebles & Xie, 2023; Ma et al., 2024; Wang et al., 2026; Karras et al., 2024; Wang et al., 2024b; Song et al., 2025; Rombach et al., 2022). However, VAE training typically involves adversarial training and perceptual supervision, complicating the overall pipeline. Insufficient VAE training can lead to decoding artifacts, limiting the broader applicability of diffusion generative models Zhou et al. (2024b). Earlier latent diffusion models primarily focused on U-Net-like architectures. Peebles & Xie (2023) introduced transformers into diffusion models, replacing the traditionally dominant U-Net. Ma et al. (2024) further validated the transformer architecture with linear flow diffusion.

**Pixel Diffusion Models**   have progressed much more slowly than their latent counterparts. Due to the vastness of pixel space, learning difficulty and computational demands are typically far greater than those of latent diffusion models (Dhariwal & Nichol, 2021; Kingma & Gao, 2023; Hoogeboom et al., 2023). Current pixel-space diffusion models still rely on long residuals (Teng et al., 2023; Dhariwal & Nichol, 2021), limiting further scaling. Early attempts primarily split the diffusion process into chunks at different resolution scales to reduce computational burdens (Chen et al., 2025b; Teng et al., 2023). Chen et al. (2025b) used the same denoising model across all scales, while Teng et al. (2023) employed distinct models for each. However, this cascaded pipeline inevitably complicates both training and sampling. Additionally, training these models at isolated resolutions hinders end-to-end optimization and requires careful design. Li et al. (2025a) constructed fractal generative models by recursively applying atomic generative modules. Zhai et al. (2024) introduces a Transformer-based normalizing flow capable of directly modeling and generating pixels.

**Generative Models with Neural Field/Function Space**   leverage neural fields to enhance VAEs (Chen et al., 2024b; Park et al., 2024), and their generative capacity still stems from a latent diffusion model. Gong et al. (2024) trains independent neural weights for each image before training a generative model on these pre-collected weights. This remains a two-stage framework and poses non-trivial challenges for large-scale training. Wang et al. (2024c) and Wang et al. (2023b) utilized coordinate encodings to enhance diffusion model performance. Beyond diffusion-based generative models, GAN-based methods (Ntavelis et al., 2022; Skorokhodov et al., 2021; Lin et al., 2019; Karras et al., 2021) also utilized neural fields or coordinate encodings.

## 3 METHOD

### 3.1 PRELIMINARY

**Diffusion Models**   gradually adds $x_0$ with Gaussian noise $\epsilon$ to perturb the corresponding known data distribution $p(x_0)$ into a simple Gaussian distribution. The discrete perturbation function of

each $t$ satisfies $\mathcal{N}(\boldsymbol{x}_t|\alpha_t\boldsymbol{x}_0, \sigma_t^2\boldsymbol{I})$, where $\alpha_t, \sigma_t > 0$. It can also be written as Equation (1):

$$\boldsymbol{x}_t = \alpha_t\boldsymbol{x}_{\text{real}} + \sigma_t\boldsymbol{\epsilon}. \tag{1}$$

Moreover, as shown in eq. (2), eq. (1) has a forward continuous-SDE description, where $f(t) = \frac{\mathrm{d}\log\alpha_t}{\mathrm{d}t}$ and $g(t) = \frac{\mathrm{d}\sigma_t^2}{\mathrm{d}t} - (\frac{\mathrm{d}\log\alpha_t}{\mathrm{d}t}\sigma_t^2)$. Anderson (1982) establishes a pivotal theorem that the forward SDE has an equivalent reverse-time diffusion process as in Equation (3), so the generating process is equivalent to solving the diffusion SDE. Typically, diffusion models employ neural networks and distinct prediction parametrization to estimate the score function $\nabla\log_x p_{\boldsymbol{x}_t}(\boldsymbol{x}_t)$ along the sampling trajectory (Song et al., 2020; Karras et al., 2022; Ho et al., 2020):

$$d\boldsymbol{x}_t = f(t)\boldsymbol{x}_t\mathrm{d}t + g(t)\mathrm{d}\boldsymbol{w}. \tag{2}$$

$$d\boldsymbol{x}_t = [f(t)\boldsymbol{x}_t - g(t)^2\nabla_{\boldsymbol{x}}\log p(\boldsymbol{x}_t)]dt + g(t)d\boldsymbol{w}. \tag{3}$$

Rectified Flow Model simplifies diffusion model under the framework of Equation (2) and Equation (3). Different from Ho et al. (2020), which introduces non-linear transition scheduling, the rectified-flow model adopts a linear function to transform data to standard Gaussian noise. Instead of estimating the score function $\nabla_{\boldsymbol{x}_t}\log p_t(\boldsymbol{x}_t)$, rectified-flow models directly learn a neural network $v_\theta(x_t, t)$ to predict the velocity field $\boldsymbol{v}_t = d\boldsymbol{x}_t = (\boldsymbol{x}_{\text{real}} - \boldsymbol{\epsilon})$.

**Diffusion Transformer** was introduced into diffusion models to replace the traditionally dominant UNet architecture (Peebles & Xie, 2023). Given a noisy image latent $\boldsymbol{x}_t$ as Equation (1), $\boldsymbol{y}$ is the condition, $t$ is the timestep, we first partition it into non-overlapping patches, converting it into a 1D sequence. These noisy patches are then processed through stacked self-attention and FFN blocks, with class label conditions incorporated via AdaLN modulation. We denote the feature of $\boldsymbol{x}_t$ as $\mathbf{X}_t$:

$$\mathbf{X}_t = \mathbf{X}_t + \text{AdaLN}(\boldsymbol{y}, \boldsymbol{t}, \text{Attention}(\mathbf{X}_t)), \tag{4}$$

$$\mathbf{X}_t = \mathbf{X}_t + \text{AdaLN}(\boldsymbol{y}, \boldsymbol{t}, \text{FFN}(\mathbf{X}_t)). \tag{5}$$

Recent architectural improvements such as SwiGLU (Touvron et al., 2023a;b), RoPE (Su et al., 2024), and RMSNorm (Touvron et al., 2023a;b) have been extensively validated in the research community (Chu et al., 2024; Yao & Wang, 2025; Lu et al., 2024; Gong et al., 2025; Gao et al., 2025; Liao et al., 2025).

**Neural Field** is usually adopted to represent a scene through MLPs that map coordinates encodings to signals (Mildenhall et al., 2021; Sitzmann et al., 2020; Park et al., 2024; Lin et al., 2019; Chen et al., 2024b). It has been widely applied to objects (Barron et al., 2021; Lombardi et al., 2019) and surface reconstruction (Wang et al., 2021; Yu et al., 2022; Wang et al., 2023a). Specifically, recall that an MLP consists of a Linear, SiLU, and another Linear, we regard $\mathbf{W}_1^n$ as the weight for the first linear layer in MLP, while $\mathbf{W}_2^n$ is for the second. If we have a neural field with a single set of MLP parameters $\boldsymbol{\theta} = \{\mathbf{W}_1, \mathbf{W}_2\}$ for naive 2D scene, given the query coordinate $(i, j)$, the coordinate will be transformed into cosine/sine encodings in Equation (6) or DCT-basis encodings in Equation (11):

$$\text{PE}(i, j) = \sin(2^0\pi i), \cos(2^0\pi i), ..., \sin(2^L\pi i), \cos(2^L\pi i), .... \sin(2^L\pi j), \cos(2^L\pi j) \tag{6}$$

Then this encoding will be fed into the neural field MLP defined by $\boldsymbol{\theta}$ to extract features $\mathbf{V}^n(i, j)$:

$$\mathbf{V}^n(i, j) = \text{MLP}_{\boldsymbol{\theta}}(\text{PE}(i, j)). \tag{7}$$

Finally, $\mathbf{V}^n(i, j)$ can be used to regress the needed value, e.g., RGB (Mildenhall et al., 2021; Sitzmann et al., 2020), density, and SDF (Yu et al., 2022).

## 3.2 DIFFUSION TRANSFORMER WITH PATCH-WISE NEURAL FIELD

While VAEs significantly reduce spatial dimensions in the latent space, rendering a single linear projection sufficient for velocity modeling in latent diffusion transformers, pixel diffusion transformers must handle substantially larger patch sizes to maintain computational parity with their latent counterparts. As shown in the right figure (Figure 3), under such conditions, a simple linear projection becomes inadequate for capturing fine details.

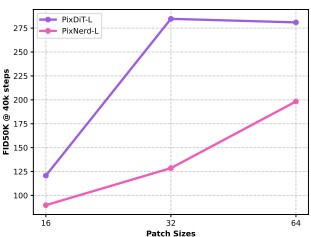

Figure 3: **Proof of Concept of Neural Field Head on Large Patch**

**Same initialization Noise with Euler-50 solver and CFG=4.0 at 400k training steps**

Figure 4: **The visualization Comparison with PixDiT-L/16 under 400k training steps.** With the help of neural field representation, our PixNerd-L/16 yields promising details and better structure.

To address the limitations of linear projection, we propose modeling patch-wise velocity decoding using an implicit neural field, which has been proven beneficial for fine details by Mildenhall et al. (2021); Sitzmann et al. (2020). Formally, given the last hidden states $\mathbf{X}^n$ (defined in Equation (4)) of the $n$-th patch in diffusion transformer, we predict neural field parameters $\boldsymbol{\theta}^n = \{\mathbf{W}_1^n \in \mathbb{R}^{D_2 \times D_1}, \mathbf{W}_2^n \in \mathbb{R}^{D_1 \times D_2}\}$ from $\mathbf{X}^n$:

$$\mathbf{W}_1^n, \mathbf{W}_2^n = \text{LINEAR}(\text{SiLU}(\mathbf{X}^n)). \tag{8}$$

To decode the pixel-wise velocity $\mathbf{v}^n(i,j)$ for the pixel coordinate $(i,j)$ in the $n$-th patch feature $\mathbf{X}^n$, where $i,j \in (0, K)$, we first encode the coordinates into encodings. These encodings ($\text{PE}(i,j)$), along with the noisy pixel value $\boldsymbol{x}^n(i,j)$), are then fed into the neural field MLP to predict the velocity. To enhance performance and stabilize training, we apply row-wise normalization to the neural field parameters:

$$\mathbf{V}^n(i,j) = \text{MLP}_{\boldsymbol{\theta}^n}(\text{CONCAT}([\text{PE}(i,j), \boldsymbol{x}^n(i,j)])), \quad \boldsymbol{\theta}^n = \{\frac{\mathbf{W}_1^n}{||\mathbf{W}_1^n||}, \frac{\mathbf{W}_2^n}{||\mathbf{W}_2^n||}\}. \tag{9}$$

Finally, shown in Equation (10) the pixel velocity $\mathbf{v}^n(i,j)$ is decoded from $\mathbf{V}^n(i,j)$ via a linear projection:

$$\mathbf{v}^n(i,j) = \text{LINEAR}(\mathbf{V}^n(i,j)). \tag{10}$$

# 4 EXPERIMENTS

We conduct ablation studies and baseline comparison experiments on ImageNet-$256 \times 256$. For class-to-image generation, we provide system-level comparisons on ImageNet-$256 \times 256$, and report FID (Heusel et al., 2017), sFID (Nash et al., 2021), IS (Salimans et al., 2016), Precision, and Recall (Kynkäänniemi et al., 2019) as the main metrics. For text-to-image generation, we report results collected on the GenEval (Ghosh et al., 2023) and DPG (Hu et al., 2024) benchmarks.

## 4.1 COMPARISON WITH BASELINES

We conduct a baseline comparison (PixDiT and PixNerd) on ImageNet $256 \times 256$. PixDiT faithfully follows the diffusion transformers (Ma et al., 2024; Peebles & Xie, 2023), using a naive linear projection to decode the velocity field. Both PixDiT-L/16 and PixNerd-L/16 are built upon SwiGLU, RoPE2d, RMSNorm, and trained with lognorm sampling. All optimizer configurations are consistently aligned. Shown in Figure 3, neural field modeling significantly improves the performance at large patch size settings. As shown in Figure 5a, our model consistently achieves lower velocity loss expectations and higher representation alignment similarity. We also provide visualization comparisons with PixDiT-L/16 in Figure 4: PixNerd-L/16, trained for the same number of steps, generates better details and structures. We provide the FID50K (without CFG) metrics of both PixDiT-L/16 and our PixNerd-L/16 in Figure 5b. PixNerd yields much better FID scores (lower is better).

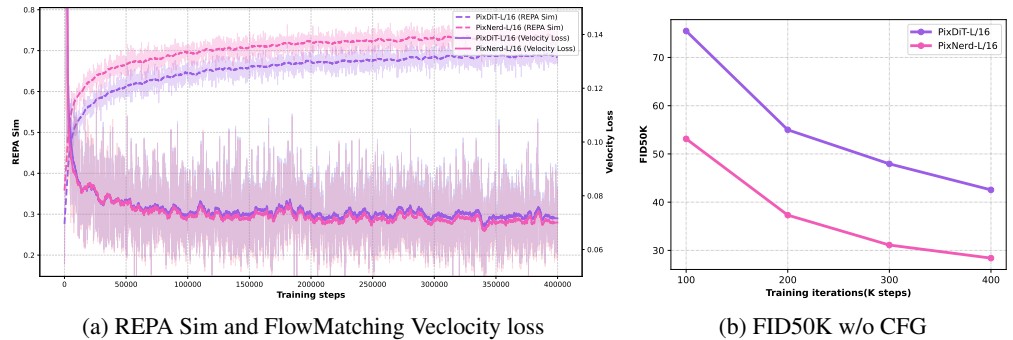

(a) REPA Sim and FlowMatching Veclocity loss

(b) FID50K w/o CFG

Figure 5: **Quantitative Comparison with PixDiT-L/16 under 400k training steps.** Our PixNerd-L/16 achieves consistently lower REPA loss and flow matching loss than its diffusion transformer counterpart.

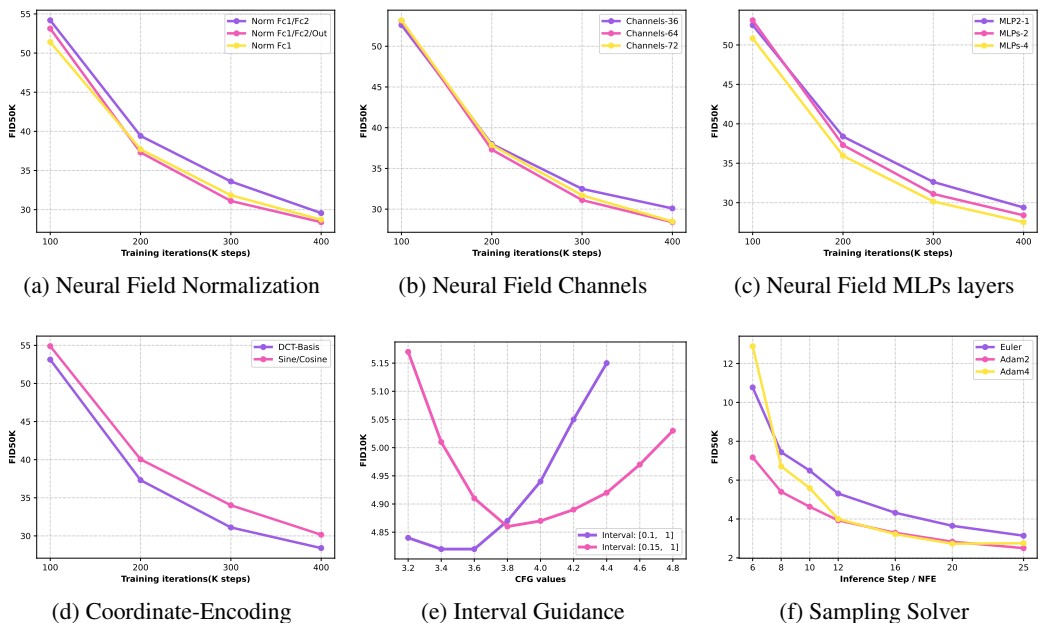

(a) Neural Field Normalization

(b) Neural Field Channels

(c) Neural Field MLPs layers

(d) Coordinate-Encoding

(e) Interval Guidance

(f) Sampling Solver

Figure 6: **Ablations studies of PixNerd.** We conduct ablation studies on class-to-image generation benchmark ImageNet256 × 256 with PixNerd-L/16.

## 4.2 NEURAL FIELD HEAD DESIGN

We conduct ablation studies on PixNerd-L/16, which comprises 22 transformer layers with 1024 channels. The Neural Field head is configured to have a computational burden comparable to that of a two-layer transformer block, so PixNerd-L/16 has inference latency similar to its counterpart, PixDiT-L/16 (24 transformer layers with 1024 channels). The detailed inference and training resource consumption can be found in the appendix.

**Neural Field Normalization**   As illustrated in Figure 6a, we evaluate different neural field normalization strategies. Our baseline compares three approaches: (1) normalizing only the first weight (FC1), (2) normalizing both weights (FC1/FC2), and (3) our default strategy that additionally normalizes the output features. Experiments demonstrate that the default strategy achieves optimal performance and convergence speed. More discussion of normalization techniques can be found in the appendix.

**Neural Field MLP Channels**   We conduct an empirical study of different MLP channel configurations (36, 64, and 72 channels) as shown in Figure 6b. Our experiments reveal that: (1) a minimal

configuration of 36 channels leads to noticeable performance degradation compared to 64 channels; (2) while the 72-channel variant achieves marginally better results, it incurs significant computational overhead, including slower training speed and increased parameter count. Based on this trade-off analysis, we select 64 channels as our default configuration.

**Neural Field MLP Depth**    We investigate the impact of neural field depth by evaluating PixNerd-L/16 with 1, 2, and 4 MLP layers, as shown in Figure 6c. Our experiments demonstrate consistent performance improvements with increasing network depth. However, considering the trade-off between computational efficiency (inference latency and training speed) and model performance, we establish the 2-layer configuration as our optimal default architecture.

**Coordinate Encodings**    We propose a novel DCT-Basis coordinate encoding as following:

$$\text{DCT-PE}(i, j) = \{\cos(k_1 i)\cos(k_2 j), \}_{k_1, k_2 \in (0, K]}. \tag{11}$$

We compare the DCT-Basis coordinate encoding (defined in Equation (11)) with traditional sine/cosine encoding in Figure 12. Our DCT-based encoding achieves significantly better results than sine/cosine encoding in terms of both convergence and final outcome. We place the implementation details of DCT-Basis encoding in the appendix.

### 4.3    INFERENCE SCHEDULER DESIGN

**Interval Guidance**    Classifier-free guidance (Peebles & Xie, 2023; Ma et al., 2024; Ho & Salimans, 2022) is a commonly used technique to improve the diffusion model performance. Interval guidance (Kynkäänniemi et al., 2024) is an improved cfg technique, and has been validated by recent works (Yao & Wang, 2025; Wang et al., 2026). We sweep different CFG values from 3.0 to 5.0 with a step of 0.2 to find the optimal CFG scheduler of PixNerd-XL/16 (800k training steps). As shown in Figure 6e, it achieves best result FID10k result with 3.4 or 3.6 within the interval $[0.1, 1]$.

**Sampling Solver**    In Figure 6f, we armed PixNerd-XL/16 with an Euler solver and an Adams-like linear multistep solver with 2/4 orders. The Adams-2 order solver consistently achieves better results than the Euler solver with limited sampling steps. Due to the learning difficulty of pixel spaces, the Adams-4-order solver performs unstably compared to the Euler and Adams2 solvers.

### 4.4    CLASS-TO-IMAGE GENERATION

**Training Details**    In class-to-image generation, to ensure a fair comparative analysis, we did not use gradient clipping and learning rate warm-up techniques. We adopt EMA with 0.9999 to stabilize the training. Our default training infrastructure consisted of $8\times$ A100 GPUs.

**Visualizations**    We placed the selected visual examples from PixNerd-XL/16 trained on ImageNet $256 \times 256$ and ImageNet $512 \times 512$ at Figure 2. Our PixNerd-XL/16 can generate images with promising details. We generate these images with a CFG of 3.5 and the Euler-50 solver.

**ImageNet** $256 \times 256$ **Benchmark**    We report the metrics of PixNerd-L/16 and PixNerd-XL/16 in Table 1. Under merely 160 training epochs, PixNerd-L/16 achieves 2.64 FID, significantly better than other pixel generative models like Jetformer (Tschannen et al., 2024), FractalMAR (Li et al., 2025a). Further, PixNerd-XL/16 achieves 2.29 FID under 50 steps with the Euler solver. When used with Adams-2-order-solver (Adam2 for brevity), PixNerd-XL/16 (800k training steps) achieves 2.16 FID, comparable to DiT. With enough sampling steps and training iterations, PixNerd-XL/16 (1600k training steps) boosts FID to 1.93 under 100 steps. We also provide PixNerd-XL/8, which achieves 1.87 FID within 800K training iterations. Moreover, our PixNerd has a much lower latency compared to other pixel space generative models.

### 4.5    TEXT-TO-IMAGE GENERATION

**Data Preprocess Details**    For text-to-image generation, we trained our model on a mixed dataset containing approximately 45M images from open-sourced datasets, e.g., SAM (Kirillov et al., 2023), JourneyDB (Sun et al., 2023), ImageNet-1K (Russakovsky et al., 2015). The detailed data source can

| | Params | Epochs | NFE | Latency(s) | FID↓ | sFID↓ | IS↑ | Pre.↑ | Rec.↑ |
|---|---|---|---|---|---|---|---|---|---|
| **ImageNet** 256×256 with CFG | | | | | | | | | |
| *Latent Generative Models* | | | | | | | | | |
| LDM-4 (Rombach et al., 2022) | 400M + 86M | 170 | 250x2 | / | 3.6 | / | 247.7 | 0.87 | 0.48 |
| DiT-XL (Peebles & Xie, 2023) | 675M + 86M | 1400 | 250x2 | 2.82 | 2.27 | 4.60 | 278.2 | 0.83 | 0.57 |
| SiT-XL (Ma et al., 2024) | 675M + 86M | 1400 | 250x2 | 2.82 | 2.06 | 4.50 | 270.3 | 0.82 | 0.59 |
| FlowDCN (Wang et al., 2024b) | 618M + 86M | 400 | 250x2 | / | 2.00 | 4.33 | 263.1 | 0.82 | 0.58 |
| REPA (Yu et al., 2024) | 675M + 86M | 800 | 250x2 | 2.82 | 1.42 | 4.70 | 305.7 | 0.80 | 0.64 |
| DDT-XL (Wang et al., 2026) | 675M + 86M | 400 | 250x2 | 2.92 | 1.26 | 4.51 | 310.6 | 0.79 | 0.65 |
| MAR-L (Li et al., 2025b) | 479M + 86M | 800 | / | / | 1.78 | / | 296.0 | 0.81 | 0.60 |
| CausalFusion (Deng et al., 2024) | 676M + 86M | 800 | 250x2 | / | 1.77 | - | 282.3 | 0.82 | 0.61 |
| *Pixel Generative Models* | | | | | | | | | |
| ADM (Dhariwal & Nichol, 2021) | 554M | 400 | 250 | 21.3 | 4.59 | 5.13 | 186.7 | **0.82** | 0.52 |
| RDM (Teng et al., 2023) | 553M + 553M | / | / | / | 1.99 | **3.99** | 260.4 | 0.81 | 0.58 |
| JetFormer (Tschannen et al., 2024) | 2.8B | / | / | / | 6.64 | / | 0.69 | 0.56 | |
| FractalMAR-H (Li et al., 2025a) | 844M | 600 | / | / | 6.15 | / | **348.9** | 0.81 | 0.46 |
| PixelFlow-XL/4 (Chen et al., 2025b) | 677M | 320 | / | 10.1 | 1.98 | 5.83 | 282.1 | 0.81 | 0.60 |
| PixDiT-L/16 (Euler-50) | 458M | 160 | 50x2 | 0.48 | 4.11 | 6.61 | 256.38 | 0.75 | 0.56 |
| **PixNerd-L/16 (Euler-50)** | 458M | 160 | 50x2 | 0.51 | 2.64 | 5.25 | 297 | 0.78 | 0.60 |
| **PixNerd-XL/16 (Adam2-50)** | 700M | 160 | 50x2 | 0.64 | 2.16 | 4.93 | 291 | 0.78 | 0.60 |
| **PixNerd-XL/16 (Euler-100)** | 700M | 160 | 100x2 | 1.28 | 2.15 | 4.55 | 297 | 0.79 | 0.59 |
| **PixNerd-XL/16 (Euler-100)** | 700M | 320 | 100x2 | 1.28 | **1.93** | 4.50 | 298 | 0.80 | **0.60** |
| PixNerd-XL/8 (Euler-100) | 700M | 160 | 100x2 | 1.98 | 1.87 | 4.36 | 298 | 0.79 | 0.61 |

Table 1: **System performance comparison** on ImageNet $256 \times 256$ class-conditioned generation. Our PixNerd-XL/16 achieves comparable results with latent diffusion models under similar computation demands while achieving much better results than other pixel space generative models. The detailed interval guidance configuration can be found in the appendix.

| | #Params | Sin.Obj. | Two.Obj. | Counting | Colors | Pos | Color.Attr. | Overall |
|---|---|---|---|---|---|---|---|---|
| | | **GenEval** Benchmark | | | | | | |
| *latent diffusion model* | | | | | | | | |
| LDM (Rombach et al., 2022) | 1.4B | 0.92 | 0.29 | 0.23 | 0.70 | 0.02 | 0.05 | 0.37 |
| DALL-E 2 | 4.2B | 0.94 | 0.66 | 0.49 | 0.77 | 0.10 | 0.19 | 0.52 |
| DALL-E 3 | - | 0.96 | 0.87 | 0.47 | 0.83 | 0.43 | 0.45 | 0.67 |
| Imagen | 3B | - | - | - | - | - | - | - |
| SD3 (Esser et al., 2024) | 8B | 0.98 | 0.84 | 0.66 | 0.74 | 0.40 | 0.43 | 0.68 |
| Transfusion (Zhou et al., 2024a) | 7.3B | - | - | - | - | - | - | 0.63 |
| *pixel diffusion model* | | | | | | | | |
| PixelFlow-XL/4 (Chen et al., 2025b) | 882M + 3B | - | - | - | - | - | - | 0.60 |
| PixelFlow-XL/4[†] (Chen et al., 2025b) | 882M + 3B | - | - | - | - | - | - | 0.64 |
| PixNerd-XXL/16 | 1.2B + 1.7B | 0.97 | 0.86 | 0.44 | 0.83 | 0.71 | 0.53 | 0.73 |

Table 2: **Comparsion with other text-to-image models on GenEval Benchmark.**[†] indicates prompt rewriting. Parameters consist of denoiser+text encoder+vae. Our PixNerd-XXL/16 achieves competitive performance compared with others under a much-limited data scale (45M images).

be found in the appendix. We recaption all the images with Qwen2.5-VL-7B (Wang et al., 2024a) to yield English caption descriptions of various lengths. Note that our caption results only contain English descriptions. All the images are cropped into a square shape of $256 \times 256$ or $512 \times 512$, we do not adopt various aspect ratio training. We leave the native resolution Wang et al. (2025) or native aspect training (Gong et al., 2025; Gao et al., 2025; Liao et al., 2025) as future work.

**Training Details** We adopt Qwen3-1.7B[1] as the text encoder. To improve the alignment of frozen text features Fan et al. (2024), we jointly train several transformer layers on the frozen text features similar to Fan et al. (2024). To further enhance the generation quality, we adopt an SFT stage at resolution $512 \times 512$ with the dataset released by Chen et al. (2025a). The total batch size is 1536 for $256 \times 256$ resolution pretraining and 512 for $512 \times 512$ resolution pretraining. We trained PixNerd on $256 \times 256$ resolution for 200K steps and trained on $512 \times 512$ resolution for 80K steps. The default training infrastructure consisted of $16 \times$ A100 GPUs. We use the Adams-2nd solver with 25 steps as the default choice for sampling.

---

[1]https://huggingface.co/Qwen/Qwen3-1.7B

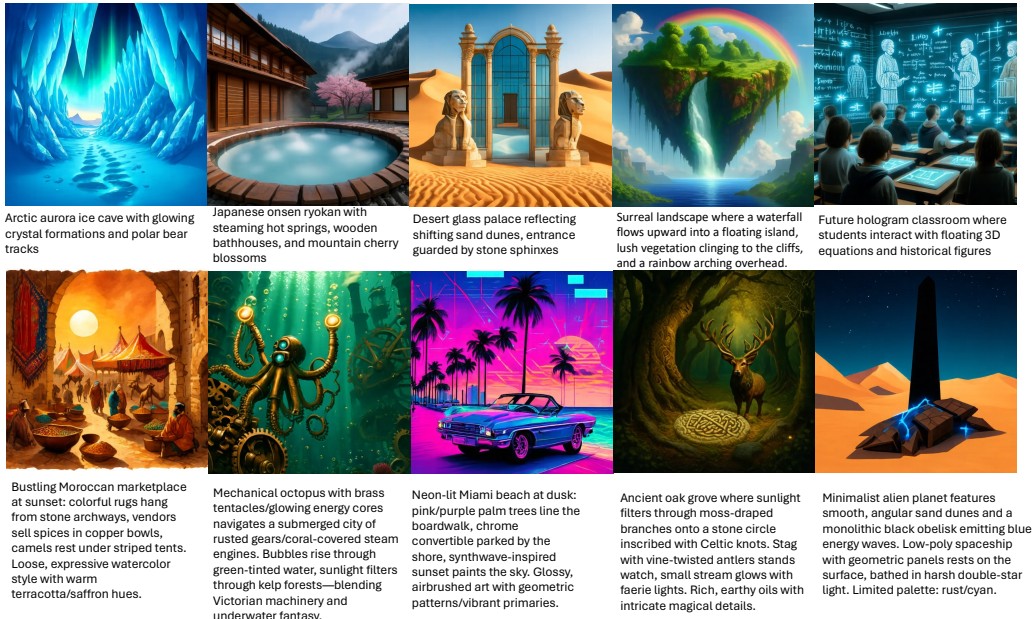

Figure 7: **The Text-to-Image** $512 \times 512$ **visualization with text descriptions of different lengths and styles.** Given text descriptions of different lengths and styles, PixNerd can generate promising samples with a large patch size of 16. We used Adams-2nd solver with 25 steps and a CFG value of 4.0 for sampling.

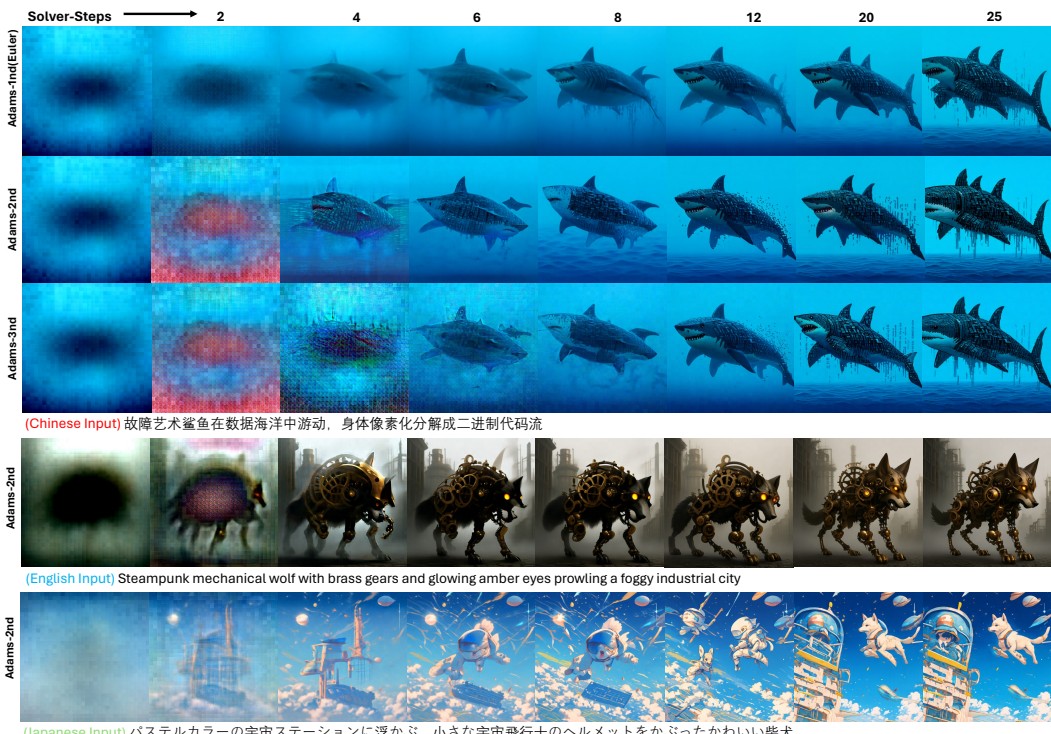

Figure 8: **The Text-to-Image** $512 \times 512$ **visualization.** We armed PixNerd with different ODE solvers, eg, Euler, Adams. Also, thanks to the powerful text embedding in Qwen3 models, though we only trained PixNerd with English captions, PixNerd can generate samples of promising quality with other languages.

| Model | #Params | DPG Benchmark | | | | | |
| | | Global | Entity | Attribute | Relation | Other | Average |
| --- | --- | --- | --- | --- | --- | --- | --- |
| *latent diffusion model* | | | | | | | |
| SD v2 | 0.86B + 0.7B+ 86M | 77.67 | 78.13 | 74.91 | 80.72 | 80.66 | 68.09 |
| PixArt-$\alpha$ | 0.61B + 4.7B + 86M | 74.97 | 79.32 | 78.60 | 82.57 | 76.96 | 71.11 |
| Playground v2 | 2.61B + 0.7B + 86M | 83.61 | 79.91 | 82.67 | 80.62 | 81.22 | 74.54 |
| DALL-E 3 | - | 90.97 | 89.61 | 88.39 | 90.58 | 89.83 | 83.50 |
| SD v1.5 | 0.86B + 0.7B + 86M | 74.63 | 74.23 | 75.39 | 73.49 | 67.81 | 63.18 |
| SDXL | 2.61B +1.4B + 86M | 83.27 | 82.43 | 80.91 | 86.76 | 80.41 | 74.65 |
| *pixel diffusion model* | | | | | | | |
| PixelFlow-XL/4 | 0.8B + 3B | - | - | - | - | - | 77.90 |
| PixNerd-XXL/16 | 1.2B + 1.7B | 80.5 | 87.9 | 87.2 | 91.3 | 72.8 | 80.9 |

Table 3: **Comparsion with other text-to-image models on DPG Benchmark.** Parameters consist of denoiser+text encoder+vae. Our PixNerd-XXL/16 achieves competitive performance compared with others under a much-limited data scale (45M images).

**Visualizations**   We provided $512 \times 512$ visualizations with prompts provided by DouBao-1.5-Pro in Figure 7. As illustrated in Figure 7, our PixNerd-XXL/16 is capable of generating visually compelling images from complex text prompts. Overall, the atmosphere and color tones are largely accurate. Noted that we only trained PixNerd with English prompts. As shown in Figure 8, we can generate images even with other languages like Chinese and Japanese thanks to the powerful embedding space of Qwen3 models (Wang et al., 2024a). We place more visual samples in the appendix.

**Sampling Solvers**   We provide denoising trajectories in $x_1$ space (clean data) of different solvers in Figure 8, including Euler, Adams-2nd, and Adams-3rd solvers. Adams-2nd solver achieves stable and fast sampling results. Thus, we take Adams-2nd as the default sampling choice.

**GenEval Benchmark**   We provided quantity comparison on Geneval (Ghosh et al., 2023) benchmark in Table 2. Our PixNerd-XXL/16 achieves comparable results under enormous patch sizes and limited data scales. As shown in Table 2, our PixNerd-XXL/16 achieves 0.73 overall score, beating Chen et al. (2025b) with a significant margin.

**DPG Benchmark**   We provided quantity comparison on DPG (Hu et al., 2024) benchmark in Table 3. Our PixNerd-XXL/16 achieves competitive results compared to its latent counterparts. As shown in Table 3, our PixNerd-XXL/16 achieves 80.9 overall score, beating other pixel generation models with a significant margin.

## 5   CONCLUSION

In this paper, we return to pixel space diffusion with neural field. We have presented a single-scale, single-stage, efficient, end-to-end solution, termed as the pixel neural field diffusion (PixNerd). Thanks to the powerful high-frequency modeling capability of the neural field head in PixNerd, we achieve very competitive results in image generation without any complex cascade design. In particular, our PixNerd-XXL/16 achieves a competitive 0.73 overall score on the GenEval benchmark and 80.9 average score on the DPG benchmark. However, pixel-space diffusion models still do not exhibit better performance scaling than advanced latent diffusion transformer models. We leave further improvements of pixel diffusion transformers to future work.

## ACKNOWLEDGMENTS

This work is partially supported by the Basic Research Program of Jiangsu (No. BK20250009) and the Collaborative Innovation Center of Novel Software Technology and Industrialization.

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

| Model | Inference | | | | Training | |
|---|---|---|---|---|---|---|
| | NFE | 1 image | 1 step | Mem (GB) | Speed (s/it) | Mem (GB) |
| SiT-L/2(VAE-f8) | 50x2 | 0.51s | 0.0097s | 2.9 | 0.30 | 18.4 |
| Baseline-L/16 | 50x2 | 0.48s | 0.0097s | 2.1 | 0.18 | 18 |
| PixNerd-L/16 | 50x2 | 0.51s | 0.010s | 2.1 | 0.19 | 22 |
| ADM-G | 50 | 4.21s | 0.08s | 2.23 | / | / |
| ADM-G | 250 | 21.3s | 0.08s | 2.23 | / | / |
| PixelFlow-XL/4 | / | 10.1s | 0.084s$^\dagger$ | 4.0 | / | / |
| PixNerd-XL/16 | 50x2 | 0.65s | 0.012s | 3.1 | 0.27 | 33.9 |

Table 4: **The resource consumption comparison.** $^\dagger$ means the average time consumption for a single step across different stages. Our PixNerd consumes much less memory and latency (Nearly $8\times$ fatser than other pixel diffusion models.

# A APPENDIX

## A.1 TRAINING AND INFERENCE COMPARSION

| | Transformer layer | Nerf MLP | Nerf PE | Nerf RGB out |
|---|---|---|---|---|
| Latency(ms) | 0.54 | 0.49 | 0.25 | 0.17 |

Table 5: **Latency of different components under Batch Size 2**

We employ **torch.compile** to optimize memory allocation and reduce redundant computations for both the baseline and PixNerd. As shown in Table 4, compared with latent counterparts, our PixNerd-L/16 achieves much higher training throughput without VAE latency and with similar inference memory. Compared to other pixel-space diffusion models, our PixNerd consumes significantly less memory and has lower latency (nearly $8\times$) faster than ADM-G and PixelFlow. We also provide detailed latency of specific layers in Table 5.

## A.2 INTERVAL GUIDANCE DETAILS OF IMAGENET 256X256

Show in Section A.2, Among the models listed, PixDiT-L/16 and PixNerd-L/16 share identical interval guidance configurations: both are trained for 160 epochs, use an interval range of 0.1 (start) to 1 (end), and adopt a CFG value of 3.5. For the PixNerd-XL/16 model, two sets of training configurations are provided: one with 160 epochs (matching the interval [0.1, 1] and CFG value 3.5 of the aforementioned two models) and another with an extended 320 epochs, where the interval remains [0.1, 1] but the CFG value is adjusted to 3.0. Finally, the PixNerd-XL/8 model is trained for 160 epochs, uses the same interval [0.1, 1] as other models, and employs a CFG value of 3.0.

## A.3 PRETRAINING DATA SOURCE OF TEXT-TO-IMAGE EXPERIMENTS

For text-to-image generation, we trained our model on a mixed dataset containing approximately 45M images from open-sourced datasets. We recaption all the images with Qwen2.5-VL-7B (Wang et al., 2024a) to yield English caption descriptions of various lengths. Note that our caption results

| Model | Epochs | Interval start | Interval end | CFG value |
|---|---|---|---|---|
| PixDiT-L/16 | 160 | 0.1 | 1 | 3.5 |
| PixNerd-L/16 | 160 | 0.1 | 1 | 3.5 |
| PixNerd-XL/16 | 160 | 0.1 | 1 | 3.5 |
| PixNerd-XL/16 | 320 | 0.1 | 1 | 3.0 |
| PixNerd-XL/8 | 160 | 0.1 | 1 | 3.0 |

Table 6: **Interval Guidance Details of Inference on ImageNet** $256 \times 256$

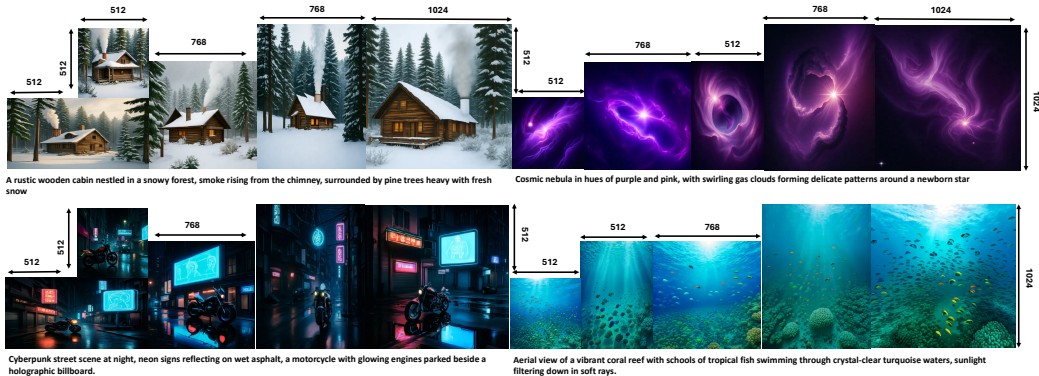

Figure 9: **Training-free arbitrary resolution generation.** We keep the amount of tokens in PixNerd as constant as pretraining resolution, we **only interpolate the neural field coordinates** for different resolutions to yield multi-resolution images.

only contain English descriptions. All the images are cropped into a square shape of $256 \times 256$ or $512 \times 512$.

Here is the detailed image source:

- ImageNet: *https://www.image-net.org/*
- JouneryDB: *https://huggingface.co/datasets/JourneyDB/JourneyDB*
- Laion(aes6+): *https://huggingface.co/datasets/laion/aesthetics_v2_4.75*
- CC12M: *https://huggingface.co/datasets/pixparse/cc12m-wds*
- SAM: *https://ai.meta.com/datasets/segment-anything*
- MidJouney23M: *https://huggingface.co/datasets/deepghs/midjourney_captioned_23m_full*

Because the webpage link is invalid in CC12M dataset and the Laion dataset, the downloaded images are incomplete, and only parts of them were obtained.

## A.4 PixNerd w/o REPA

PixNerd is training with REPA as a default configuration. We also provide ablation w/o REPA here. We trained the model for 200K steps under batch size 256 on ImageNet256.

| | FID | SFID | IS | Precision | Recall |
|---|---|---|---|---|---|
| w/o REPA | 67.68 | 9.96 | 19.41 | 0.338 | 0.551 |
| REPA loss weight 0.1 | 42.49 | 10.19 | 37.11 | 0.448 | 0.61 |
| **REPA loss weight 0.5** | 37.31 | 10.44 | 42.92 | 0.426 | 0.6275 |

Table 7: **Performance metrics under different REPA configurations** The default REPA loss weight achieves better performance.

## A.5 Arbitrary Resolution Inference

As shown in Figure 9, without any resolution adaptation fine-tuning, we can achieve arbitrary resolution generation through the coordinate interpolation while keeping the amount of tokens as constant as the pretraining resolution. Specifically, we sampled pretraining resolution from the given noisy image, then fed this sampled version into the transformer. This keeps the token amount in our PixNerd as consistent as in the pretraining stage. To match the velocity field with the desired resolution of the given noisy image, we then interpolate the coordinates for neural field decoding accordingly.

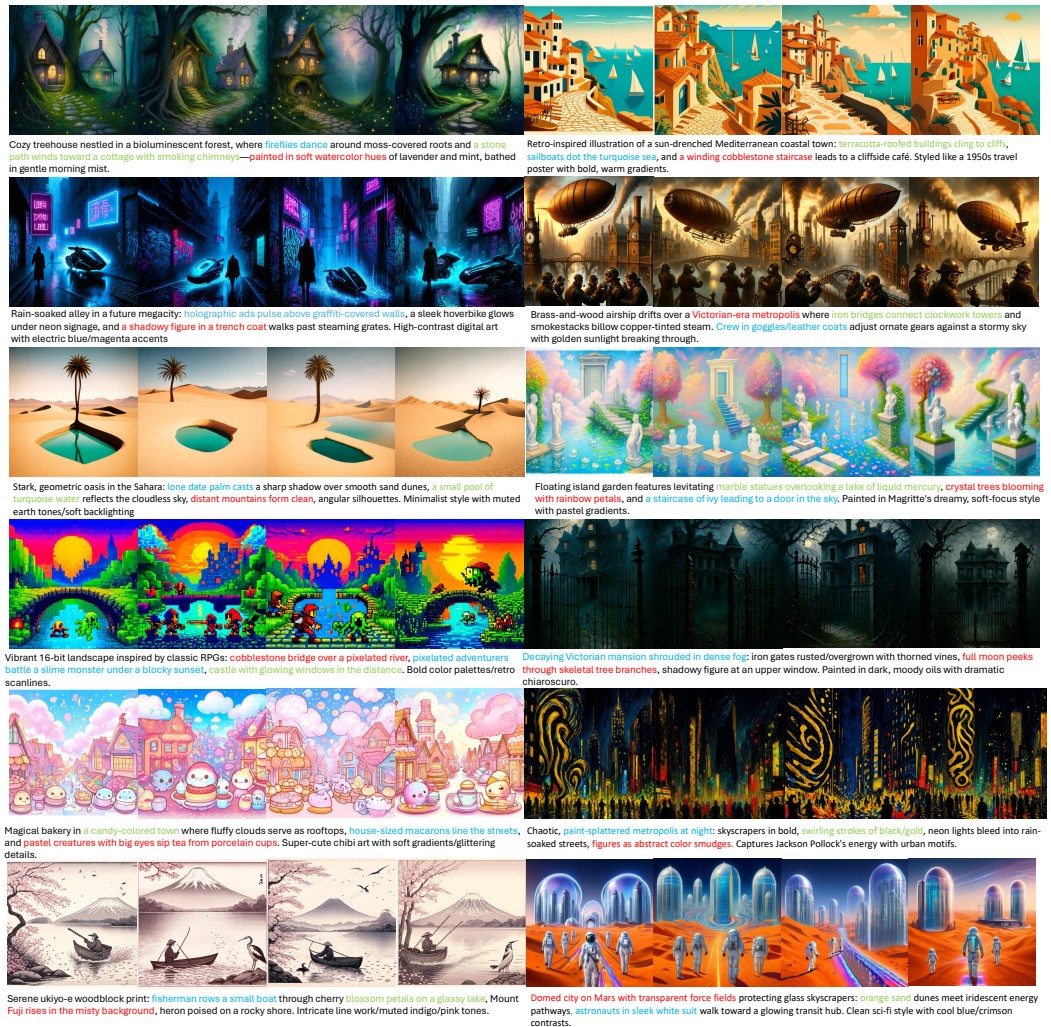

Figure 10: **The Text-to-Image** $512 \times 512$ **visualization with different initialization noises.** Given different initialization noises, our PixNerd can produce samples with correct semantics and diverse styles. This indicates a superior generation diversity and semantic following capability.

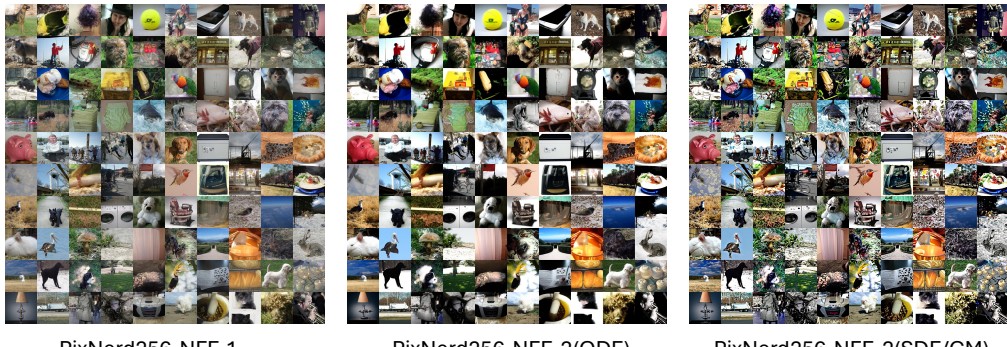

PixNerd256-NFE-1        PixNerd256-NFE-2(ODE)        PixNerd256-NFE-2(SDE/CM)

Figure 11: **The visual samples from Distilled-PixNerd-XL/16.** We employed CM sampler and ODE sampler for few-steps inference.

## A.6 MORE VISUALIZATIONS OF PIXNERD-T2I

Occasional blurry or unnatural artifacts appear in certain scenarios (e.g., steampunk lab image). We posit that appropriate post-training processing could mitigate such artifacts Ren et al. (2024), and we intend to explore pixel-space post-training as future work.

## A.7 FEW STEP GENERATION OF PIXNERD.

| Model | FID↓ | NFE↓ | #Params |
|---|---|---|---|
| *GAN Models* | | | |
| StyleGAN-XL (Sauer et al., 2022) | 2.30 | - | - |
| PagoDA (Kim et al., 2024)† | 1.56 | - | - |
| *Latent Space Diffusion Models* | | | |
| iCT-XL/2 (Dao et al., 2025) | 34.24 | 1+VAE | 84M + 675M |
| | 20.30 | 2+VAE | 84M + 675M |
| Shortcut-XL/2 (Frans et al., 2024) | 10.60 | 1+VAE | 84M + 675M |
| | 7.80 | 4+VAE | 84M + 675M |
| IMM (Zhou et al., 2025) | 8.05 | 1+VAE | 84M + 675M |
| *Pixel Space Diffusion Model* | | | |
| **Distilled-PixNerd-XL/16** | 6.76 | 1 | 700M |
| **Distilled-PixNerd-XL/16** | **4.56** | 2 | 700M |

Table 8: **Few-step Generation Performance of PixNerd.** PixNerd-XL/16 achieves superior performance on few-step generation. † emploied a multi-stage architecture and initialized from diffusion models.

Currently, however, our focus is on improving the upper-bound performance of pixel diffusion while maintaining latency comparable to latent diffusion models. Having said that, PixNerd extends classic DiT/SiT architectures with minimal modifications to enhance performance, thus maintaining compatibility with existing few-step distillation techniques. We reserve the exploration of distillation-based enhancements for future work. With limited training and parameter tuning, we achieved an FID of 4.56 at 2 NFE and 6.76 at 1 NFE within 10 epochs of finetuning. We also provide visual samples from Distilled-PixNerd-XL/16 in Figure 11.

## A.8 CLASS-TO-IMAGE IMAGENET 512x512 BENCHMARK

We provide the final metrics of PixNerd-XL/16 at Table 9. To validate the superiority of our PixNerd model, we take our PixNerd-XL/16 trained on ImageNet $256 \times 256$ as the initialization, fine-tune our PixNerd-XL/16 on ImageNet $512 \times 512$ for 200K steps. We adopt the aforementioned interval

| | ImageNet $512 \times 512$ | | | | | |
|---|---|---|---|---|---|---|
| Model | Params | FID↓ | sFID↓ | IS↑ | Pre.↑ | Rec.↑ |
| *Latent Diffusion Models* | | | | | | |
| DiT-XL/2 (Peebles & Xie, 2023) | 675M + 86M | 3.04 | 5.02 | 240.82 | 0.84 | 0.54 |
| SiT-XL/2 (Ma et al., 2024) | 675M + 86M | 2.62 | 4.18 | 252.21 | 0.84 | 0.57 |
| REPA-XL/2 (Yu et al., 2024) | 675M + 86M | 2.08 | 4.19 | 274.6 | 0.83 | 0.58 |
| FlowDCN-XL/2 (Wang et al., 2024b) | 608M + 86M | 2.44 | 4.53 | 252.8 | 0.84 | 0.54 |
| EDM2 (Karras et al., 2022) | 1.5B + 86M | 1.81 | | | | |
| DDT-XL/2 (Wang et al., 2026) | 675M + 86M | 1.28 | 4.22 | 305.1 | 0.80 | 0.63 |
| *Pixel Diffusion Models* | | | | | | |
| ADM-G (Dhariwal & Nichol, 2021) | 559M | 7.72 | 6.57 | 172.71 | 0.87 | 0.42 |
| ADM-G, ADM-U | 559M | 3.85 | 5.86 | 221.72 | 0.84 | 0.53 |
| RIN (Jabri et al., 2022) | 320M | 3.95 | - | 210 | - | - |
| SimpleDiffusion (Hoogeboom et al., 2023) | 2B | 3.54 | - | 205 | - | - |
| **PixNerd-XL/16 (100K+CFG 3.5)** | 700M | 2.84 | 5.95 | 245.62 | 0.80 | 0.59 |
| **PixNerd-XL/16 (100K+CFG 4.5)** | 700M | 2.56 | 5.50 | 277.80 | 0.80 | 0.60 |
| **PixNerd-XL/16 (200K+CFG 4.5)** | 700M | 2.45 | 5.19 | 275.77 | 0.79 | 0.63 |

Table 9: **Benchmarking class-conditional image generation on ImageNet $512 \times 512$.** Our PixNerd-XL/16($512 \times 512$) is fine-tuned from the same model trained on $256 \times 256$ resolution for 200K steps. We adopt the interval guidance with interval $[0.1, 1]$ and CFG of 4.5. We take Euler solver with 100 steps as the default choice.

guidance Kynkäänniemi et al. (2024) and we achieved 2.45 FID, with CFG of 4.5 within the time interval $[0.1, 1.0]$. PixNerd-XL/16 achieves comparable performance to other diffusion models.

## A.9 THE DETAILS OF NEURAL FIELD HEAD

As shown in Equation (9), we normalize the predicted neural field parameters $W_1$, $W_2$, and the final feature of the neural field(Out) as the default configuration. Specifically, given the predicted weight matrix $W = [w_1, w_2, ...w_n]^T \in R^{D_1 \times D_2}$, we normalize $W$ row-wisely as $W = [\frac{w_1}{||w_1||}, \frac{w_2}{||w_2||}, ... \frac{w_2}{||w_2||}]^T \in R^{D_1 \times D_2}$.

```python
class NerfBlock(nn.Module):
    def __init__(self, hidden_size_s, hidden_size_x, mlp_ratio=4):
        super().__init__()
        self.param_generator1 = nn.Sequential(
            nn.Linear(hidden_size_s, 2*hidden_size_x**2*mlp_ratio, bias=
                True),
        )
        self.norm = RMSNorm(hidden_size_x, eps=1e-6)
        self.mlp_ratio = mlp_ratio
    def forward(self, x, s):
        batch_size, num_x, hidden_size_x = x.shape
        mlp_params1 = self.param_generator1(s)
        fc1_param1, fc2_param1 = mlp_params1.chunk(2, dim=-1)
        fc1_param1 = fc1_param1.view(batch_size, hidden_size_x,
            hidden_size_x*self.mlp_ratio)
        fc2_param1 = fc2_param1.view(batch_size, hidden_size_x*self.
            mlp_ratio, hidden_size_x)

        # normalize fc1
        normalized_fc1_param1 = torch.nn.functional.normalize(fc1_param1,
            dim=-2)
        # normalize fc2 (optional for T2I Task)
        normalized_fc2_param1 = torch.nn.functional.normalize(fc2_param1,
            dim=-2)
        # mlp 1
        res_x = x
```

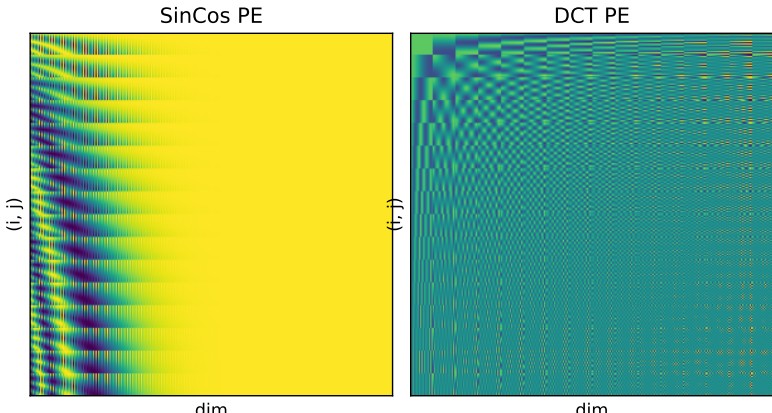

Figure 12: **DCT PE and Classic Sin/Cos PE.** DCT PE is more sensitive to position changes than Classic Sin/Cos PE.

```
22          x = self.norm(x)
23          x = torch.bmm(x, normalized_fc1_param1)
24          x = torch.nn.functional.silu(x)
25          x = torch.bmm(x, normalized_fc2_param1)
26          x = x + res_x
27          return x
```

### A.10    THE DETAILS DCT ENCODINGS

We provide the feature visualization of DCT-Based position encoding and classic sin/cos position encoding in Figure 12. Shown in Figure 12, DCT PE is more sensitive to position changes. Below, we provide the detailed code implementation of DCT PE.

```
1     def fetch_pos(self, patch_size, device, dtype):
2         pos_x = torch.linspace(0, 1, patch_size, device=device, dtype=
              dtype)
3         pos_y = torch.linspace(0, 1, patch_size, device=device, dtype=
              dtype)
4         pos_y, pos_x = torch.meshgrid(pos_y, pos_x, indexing="ij")
5         pos_x = pos_x.reshape(-1, 1, 1)
6         pos_y = pos_y.reshape(-1, 1, 1)
7
8         freqs = torch.linspace(0, self.max_freqs, self.max_freqs, dtype=
              dtype, device=device)
9         freqs_x = freqs[None, :, None]
10        freqs_y = freqs[None, None, :]
11        coeffs = (1 + freqs_x * freqs_y) ** -1
12        dct_x = torch.cos(pos_x * freqs_x * torch.pi)
13        dct_y = torch.cos(pos_y * freqs_y * torch.pi)
14        dct = (dct_x * dct_y * coeffs).view(1, -1, self.max_freqs ** 2)
15        return dct
```

### A.11    THE RELATION BEHIND NEURAL FIELD NORMALIZATION AND QK-NORM

Generally, QK-Norm plays a crucial role in modern transformers. To better understand the weight normalization, we can imagine the neural field decoding as a cross-attention. Coordinate encodings in a local patch look like queries, while patch-wise $W_1$ is the keys, and patch-wise $W_2$ is the values. This analogy can also explain why only normalizing $W_1$ in Equation (9) delivers comparable results (in Figure 6a) in limited training iterations(up to 400k steps). Our further experiments on text-to-image also reveal that only normalizing $W_1$ is sufficient.

However, since class-to-image models are typically trained with far more steps than text-to-image models (1.6M steps vs. 200k steps), we observe that normalizing only $W_1$ leads to loss spikes in the **class-to-image** task. Even though these spikes do not crash down the training, we still prune to normalize all fully connected (FC) layers to ensure training stability.

## A.12 DISCUSSION WITH OTHER RELATED WORKS

Several related works (Chen et al., 2024b; Park et al., 2024; Wang et al., 2024c) also combine diffusion with neural fields. In practice, however, they differ fundamentally from PixNerd. For example, INFD (Chen et al., 2024b) and DDMI (Park et al., 2024) leverage neural fields to enhance VAEs rather than diffusion models, and their generative capacity still stems from a latent diffusion model. DenoisedWeights (Gong et al., 2024) trains independent neural weights for each image before training a generative model on these pre-collected weights. This remains a two-stage framework and poses non-trivial challenges for large-scale training. INRFlow Wang et al. (2024c) and PatchDiffusion(Wang et al., 2023b) utilize coordinate encodings to enhance diffusion model performance. Beyond diffusion-based generative models, GAN-based methods (Ntavelis et al., 2022; Skorokhodov et al., 2021; Lin et al., 2019; Karras et al., 2021) also utilize neural fields or coordinate encodings. PixNerd is a simple yet elegant single-stage pixel-space generative model that does not rely on a VAE. Current latent generative models inevitably cascade errors due to their two-stage configurations. Further, a high-quality VAE usually demands numerous losses supervisions, e.g., adversarial loss, LPIPS loss. In particular, adversarial loss is unstable in training and tends to introduce artifacts. Pixel generative model has more potential in the future, and PixNerd is a simple yet elegant solution for a pixel-space generative model.

## A.13 LARGE LANGUAGE MODEL USAGE

We only use LLM to polish sentence-level writing and grammar. We confirm that the revised sentences fully reflect our original texts.

