# OpenReview forum: "PixNerd: Pixel Neural Field Diffusion"
_ICLR.cc/2026/Conference — ICLR 2026 Poster_

### Official Review · Reviewer_Aqjg · 2025-10-27

**Soundness:** 4
**Presentation:** 2
**Contribution:** 3
**Rating:** 8
**Confidence:** 5

**Summary:**

This paper introduces PixNerd, a single-stage, end-to-end diffusion transformer that operates directly in pixel space. It aims to solve the trade-off between VAE-based latent models (which brings 2-stage training and is not elegant) and traditional pixel-space models (which are computationally expensive or rely on complex cascades).

The core technical contribution is a novel decoder. Instead of using a simple linear projection on large patches (which loses detail), the diffusion transformer's final features are used to predict the weights of a small MLP (a "neural field head") for each patch. This patch-specific MLP then decodes the final velocity for each pixel within that patch by taking the pixel's local coordinates as input. This approach allows the model to be computationally efficient (using large patches) while retaining high-frequency detail (via the neural field decoder).

**Strengths:**

This paper proposes an alternative approach besides latent diffusion models. It has several strengths:

1. Elimination of VAE: The single-stage, end-to-end pipeline avoids the pre-trained VAE, removing its associated decoding artifacts and 2-stage training.

2. Efficient Pixel-Space Modeling: The proposed "neural field head" is an effective method for combining the computational efficiency of large-patch transformers with the high-fidelity representation of neural fields.

3. Strong Benchmark Performance: Achieves a 1.93 FID on ImageNet 256x256, which is state-of-the-art for pixel-space models and 3. competitive with top-tier latent-space models. It also demonstrates superior performance on text-to-image generation tasks.

4. Flexible Resolution: The coordinate-based nature of the decoder allows for training-free scaling to arbitrary resolutions at inference time, a significant practical benefit.

**Weaknesses:**

1. New Artifacts: While VAE artifacts are removed, the authors note in the appendix (A.4) that the model can introduce its own "blurry or unnatural artifacts."

2. Architectural Complexity: The design trades the complexity of a VAE for the complexity of the neural field head. Ablation studies (4.2) show performance is sensitive to this head's depth, channel count, and normalization strategy.

3. Potential Training Instability: The paper discusses the necessity of specific normalization strategies (Fig 5a, A.9) to "ensure training stability" and prevent "loss spikes" during long training runs, suggesting a sensitive training dynamic.

4. The main insight of this paper is to use NeRF MLP Head to replace the traditional MLP Head. Does it induce some specific designs other than the original NeRF method?

**Questions:**

1. Why does PixNerd have lower latency than the traditional latent diffusion models? Any insights here?

2. On text-to-image generation task, the parameters of PixNerd is 1.2B + 1.7B. What modules do these two parameters correspond to?

3. For NeRF MLP depth ablation in figure 5 (c), does the total number of layers is the same? For PixNerd-L/16 with 4 MLP layers, whether it use 22 transformer layers or it use 20 transformer layers? Further, given the same total layer number, which is the best ratio for transformer layers and NeRF MLP layers?

4. How about the latency comparison of transformer layers and NeRF MLP layers?

5. Based on figure 5 (d) DCT-Basis is better than Sin/Cos coordinate-encoding. Any insights for this?

---

> ### Author Response · Authors · 2025-11-17
>
> **Q.1 Why does PixNerd have lower latency than the traditional latent diffusion models?**
>
> We configured such that a NeRF MLP layer exhibits similar latency to a transformer layer in PixNerd. Consequently, both computational complexity and time cost are largely consistent with those of the standard DIT for 1-NFE. In Table.1 and Table.4, we provide the detailed inference configuration. PixNerd achieves 1.93 FID with **100 steps** (2x100 NFE for CFG), while latent counterparts usually employ **250 steps** (2x250 NFE for CFG). Also, PixNerd **does not need VAE**, elimiating the decoding latency of VAE decoder.
>
> **Q.2. On text-to-image generation task, the parameters of PixNerd is 1.2B + 1.7B. What modules do these two parameters correspond to?**
>
> 1.2B refers to the PixNerd diffusion parameters, 1.7B refers to the text encoder(Qwen3-1.7B)
>
> **Q.3 For NeRF MLP depth ablation in figure 5 (c), does the total number of layers is the same?**
>
> No, we keep the number of transformer layers as consistent, varying the number of Nerf MLPs.
>
> **Q.4 The best ratio for transformer layers and NeRF MLP layers.**
>
> We provide the detailed metrics of the different layer ratios here under 200K training steps. The default 22:2 ratio achieves the best performance than others.
>
> | Transformer layers | nerf | FID   | SFID  | IS    | Recall   | Precision |
> | ----------------- | ---- | ----- | ----- | ----- | -------- | --------- |
> | 23                | 1    | 39.40 | 10.49 | 41.68 | 0.45     | 0.628     |
> | 22   **(default)**             | 2    | 37.31 | 10.44 | 42.92 | 0.426    | 0.6275    |
> | 20                | 4    | 39.43 | 11.0  | 40.49 | 0.45172  | 0.6261    |
> | 22                | 1    | 38.39 | 10.26 | 42.96 | 0.45     | 0.62      |
> | 22                | 4    | 35.95 | 10.99 | 44.11 | 0.47     | 0.636     |
>
> **Q.5 the latency comparison of transformer layers and NeRF MLP layers.**
>
> We configured such that a NeRF MLP layer exhibits similar latency to a transformer layer in PixNerd. We will provide detailed latency here.
>
> |Bsz2 |Transformer single layer|Nerf MLP|Nerf PE|	Nerf RGB out|
> |-----------|---------|---------|----------|----------|
> |latency(ms)|	0.54|	0.49|	0.25|	0.17|
>
> **Q.6 Potential Training Instability.**
>
> In Appendix 9, we discuss the normalization strategies. Though "loss spikes" occur during long training runs without normalization, this spike does not harm the performance and can be avoided by normalization. Thus our PixNerd is relatively stable and validated in the text-to-image applications.
>
> **Q.7 DCT-Basis is better than Sin/Cos coordinate-encoding**
>
> DCT is more sensitive to position changes than SinCos. We provide a visual comparison: https://anonymous.4open.science/r/PixNerd_Rebuttal-1EF8/dct_pe.pdf

---

### Official Review · Reviewer_vPnv · 2025-11-02

**Soundness:** 4
**Presentation:** 4
**Contribution:** 4
**Rating:** 8
**Confidence:** 3

**Summary:**

This paper argues that current latent diffusion models are limited the VAE which introduces encoding and decoding errors. This paper proposes to directly model pixel space with large patch sizes and learn neural fields to decode large patches. The new approach avoids the two-stage training process of the latent diffusion. The paper reports state-of-the-art results on ImageNet 256x256 with a FID score of 1.93 and a nearly 8x reduction in latency compared to existing methods. The authors also demonstrate the model's effectiveness in text-to-image generation, achieving competitive scores on the GenEval and DPG benchmarks.

**Strengths:**

- Novel method. The proposed PixNerd architecture, which combines a large-patch diffusion transformer with a neural field decoder, is a novel and interesting approach to pixel-space diffusion modeling.

- Strong results. This paper reported a competitive FID score of 1.93 on ImageNet. Moreover, the proposed framework can be applied to text-to-image generation and achieves a competitive 0.73 overall score on the GenEval benchmark and 80.9 overall score on the DPG benchmark.

- Efficiency. This paper shows a significant latency reduction compared to both pixel diffusion and latent diffusion methods.

**Weaknesses:**

- While the results on ImageNet at 256x256 resolution are quite competitive (see Table 1), the results at 512 resolution are not so convincing (see Table 6). The authors are encouraged to explain the performance degradation. The authors are also encouraged to provide comparisons at a even higher resolution like 1024 or 768.

-Unclear latency comparison. The abstract mentions an 8x latency improvement but does not specify which models were used for comparison. According to Table 1, PixNerd did not achieve 8x latency improvement compared to latent diffusion methods.

**Questions:**

How does the performance of PixNerd, in terms of both image quality and latency, scale as the image resolution increases (e.g., 512x512, 1024x1024)?

---

> ### Author Response · Authors · 2025-11-17
>
> **Q.1 The inferior performance of PixNerd on 512 resolution.**
>
> We attribute the suboptimal performance at 512 resolution to limited training steps, as PixNerd was only finetuned at this resolution for 100k steps. We resume training from 100K steps to 200K steps, and adjust the CFG value from 3.5 to 4.5. Our PixNerd-XL/16 achieves 2.45 FID performance.
>
> |  |  | **ImageNet 512×512** |  |  |  |  |
> |------------------------------|---------|----------------|----------------|----------------|----------------|----------------|
> | Model | Params | FID↓ | sFID↓ | IS↑ | Pre.↑ | Rec.↑ |
> | **Latent Diffusion Models** |  |  |  |  |  |  |
> | DiT-XL/2 | 675M + 86M | 3.04 | 5.02 | 240.82 | 0.84 | 0.54 |
> | SiT-XL/2 | 675M + 86M | 2.62 | 4.18 | 252.21 | 0.84 | 0.57 |
> | FlowDCN-XL/2 | 608M + 86M | 2.44 | 4.53 | 252.8 | 0.84 | 0.54 |
> | **Pixel Diffusion Models** |  |  |  |  |  |  |
> | ADM-G  | 559M | 7.72 | 6.57 | 172.71 | 0.87 | 0.42 |
> | ADM-G, ADM-U | 559M | 3.85 | 5.86 | 221.72 | 0.84 | 0.53 |
> | RIN | 320M | 3.95 | - | 210 | - | - |
> | SimpleDiffusion | 2B | 3.54 | - | 205 | - | - |
> | **PixNerd-XL/16 (FT100K+CFG 3.5)** | 700M | 2.84 | 5.95 | 245.62 | 0.80 | 0.59 |
> | **PixNerd-XL/16 (FT100K+CFG 4.5)** | 700M | 2.56 | 5.50 | 277.80 | 0.80 | 0.60 |
> | **PixNerd-XL/16 (FT200K+CFG 4.5)** | 700M | 2.45 | 5.19 | 275.77 | 0.79 | 0.63 |
>
>
> **Q.2 The image quality and latency of PixNerd on other resolutions**
>
> We configured that one Nerf MLP layer will have similar latency as a transformer layer of the PixNerd. Thus, both the computational complexity and time cost are basically consistent with those of the standard DIT. We will benchmark our PixNerd on different resolutions and add these to the revised version. We provide sample visualization at: https://anonymous.4open.science/r/PixNerd_Rebuttal-1EF8/pixnerd_256_vs_512.pdf
>
> | Resolution | 256x256 | 512x512 | 1024x1024 |
> | ---- | ---- | ---- | ---- |
> | bs2 | 0.012s | 0.024s | 0.099s |
> | bs32 | 0.07s | 0.40s | 1.45s |

---

### Official Review · Reviewer_iKuE · 2025-11-03

**Soundness:** 3
**Presentation:** 3
**Contribution:** 3
**Rating:** 4
**Confidence:** 4

**Summary:**

The paper introduces PixNerd, a novel end-to-end pixel-space diffusion transformer that replaces the traditional linear projection head with a neural field decoder. Its main goal is to bring the efficiency and visual fidelity of latent diffusion transformers into the raw pixel domain, without relying on VAEs or multi-scale cascades that introduce complexity and latency.

**Strengths:**

S1. Originality and Significance

- I highly commend the paper’s central motivation and direction. While most recent efforts focus on reducing computational cost by operating entirely in latent space, this work takes the opposite yet equally important perspective to explore how to lower cost directly in pixel space. This inversion of the conventional design philosophy is both insightful and original, addressing a long-standing challenge in diffusion modeling.

S2. Experimental Quality and Clarity

- I also deeply appreciate the thorough and well-executed ablation studies. The paper demonstrates an exceptional level of experimental rigor, allowing readers to form a fair and comprehensive understanding of the proposed method’s behavior. In my view, this kind of meticulous empirical investigation exemplifies what a well-written paper should strive for.

**Weaknesses:**

W1. Justification for architectural necessity.

- Given that recent advances in diffusion distillation (e.g., one-step or few-step distilled diffusion models) can achieve nearly identical performance to full diffusion models with drastically reduced sampling steps, it is unclear why PixNerd needs to exist as a separate model. Could the authors clarify whether PixNerd itself can serve as a teacher model for distillation? If not, then PixNerd would likely exhibit much higher latency compared to diffusion + distillation pipelines. Moreover, the claimed 8× speedup still appears slower than VAE + diffusion pipelines, which challenges the practical advantage of the method. If the authors could convincingly argue in this point, then I will flip my assessment.

W2. Missing baselines and quantitative comparison.

- In Table 1, it would significantly strengthen the evaluation to include representative generative baselines such as StyleGAN [1], CDM [2], Simple Diffusion [3], VDM++ [4], and PaGoDA [5].
In particular, the paper should provide quantitative comparisons with one-step models (either GAN or diffusion + distillation) and explicitly report latency differences to position PixNerd more clearly within the current landscape of efficient generative models.

[1] Sauer, Axel, Katja Schwarz, and Andreas Geiger. "Stylegan-xl: Scaling stylegan to large diverse datasets." ACM SIGGRAPH 2022 conference proceedings. 2022.
[2] Ho, Jonathan, et al. "Cascaded diffusion models for high fidelity image generation." Journal of Machine Learning Research 23.47 (2022): 1-33.
[3] Hoogeboom, Emiel, Jonathan Heek, and Tim Salimans. "simple diffusion: End-to-end diffusion for high resolution images." International Conference on Machine Learning. PMLR, 2023.
[4] Kingma, Diederik, and Ruiqi Gao. "Understanding diffusion objectives as the elbo with simple data augmentation." Advances in Neural Information Processing Systems 36 (2023): 65484-65516.
[5] Kim, Dongjun, et al. "Pagoda: Progressive growing of a one-step generator from a low-resolution diffusion teacher." Advances in Neural Information Processing Systems 37 (2024): 19167-19208.

**Questions:**

-

---

> ### Author Response · Authors · 2025-11-17
>
> **Q.1   Architectural necessity of PixNerd.**
>
> Current diffusion models[1, 2] boast a larger parameter count than VAEs, which calls into question the necessity of VAEs altogether. This motivates our return to the pixel space. However, under comparable computational resources to latent diffusion models, vanilla diffusion transformers struggle to efficiently model high-frequency details directly in the pixel space. This motivates us to propose PixNerd with minimal modifications to SiT/DiT.  Currently, pixel-space diffusion models are relatively underdeveloped, so we still focus on performance improvement rather than acceleration; we leave few-step exploration as future work(See Q.3).
>
> **Q.2 The comparison ambiguity(Unclear latency comparison)**
>
> As pointed by reviewer@vPnv, in the abstract, *'nearly 8x lower latency without any complex cascade pipeline or VAE'* indeed has ambiguity,  as the compared counterparts are pixel diffusions like classic ADM, and recent PixelFlow. We achieved much better results with much lower latency than previous pixel diffusion models without a cascade pipeline and VAE.  We provide detailed latency in Table.1 and Table.4.  We will revise the abstract with more precious statements.
>
> **Q.3 PixNerd with Distillation.**
>
> Currently, however, our focus is on improving the upper-bound performance of pixel diffusion while maintaining latency comparable to latent diffusion models.  Having said that, PixNerd extends classic DiT/SiT architectures with minimal modifications to enhance performance, thus maintaining compatibility with existing few-step distillation techniques. We reserve the exploration of distillation-based enhancements for future work. Due to time constraints, we could only provide very preliminary experimental results during the rebuttal period. With limited training and parameter tuning, we achieved an FID of 4.56 at 2 NFE and 6.76 at 1 NFE within 10 epochs of finetuning.  We provide the sample visualization at: https://anonymous.4open.science/r/PixNerd_Rebuttal-1EF8/pixenerd_few_steps.pdf
>
> | Model | FID$\downarrow$ | NFE$\downarrow$ | \#Params |
> | --- | --- | --- | --- |
> | **Models in Latent Space** |  |  |  |
> | iCT-XL/2 | 34.24 | 1+VAE | 84M + 675M |
> |  | 20.30 | 2+VAE | 84M + 675M |
> | Shortcut-XL/2 | 10.60 | 1+VAE | 84M + 675M |
> |  | 7.80 | 4+VAE | 84M + 675M |
> | IMM | 8.05 | 1+VAE | 84M + 675M |
> | **Models in Pixel Space** |  |  |  |
> | **Distilled-PixNerd-XL/16** | 6.76 | 1 | 700M |
> | **Distilled-PixNerd-XL/16** | **4.56** | 2 | 700M |
>
> *Table: **Few-step Generation Performance of PixNerd.** PixNerd-XL/16 achieves superior performance on few-step generation.*
>
> **Q.4 Missing baselines and quantitative comparison.**
>
> We will revise our manuscript and add these missing baselines[3,4,5,6,7] to a new distillation section.
>
> [1] Wu, Chenfei, et al. "Qwen-image technical report." arXiv preprint arXiv:2508.02324 (2025).
>
> [2] Esser, Patrick, et al. "Scaling rectified flow transformers for high-resolution image synthesis." Forty-first international conference on machine learning. 2024.
>
> [3] Sauer, Axel, Katja Schwarz, and Andreas Geiger. "Stylegan-xl: Scaling stylegan to large diverse datasets." ACM SIGGRAPH 2022 conference proceedings. 2022.
>
> [4] Ho, Jonathan, et al. "Cascaded diffusion models for high fidelity image generation." Journal of Machine Learning Research 23.47 (2022): 1-33.
>
> [5] Hoogeboom, Emiel, Jonathan Heek, and Tim Salimans. "simple diffusion: End-to-end diffusion for high resolution images." International Conference on Machine Learning. PMLR, 2023.
>
> [6] Kingma, Diederik, and Ruiqi Gao. "Understanding diffusion objectives as the elbo with simple data augmentation." Advances in Neural Information Processing Systems 36 (2023): 65484-65516.
>
> [7] Kim, Dongjun, et al. "Pagoda: Progressive growing of a one-step generator from a low-resolution diffusion teacher." Advances in Neural Information Processing Systems 37 (2024): 19167-19208.

---

> ### Author Response · Authors · 2025-11-26
>
> Dear Reviewer@iKuE, thank you for your thorough evaluation of our manuscript and your valuable feedback. As the ICLR rebuttal period is nearing its end, we have addressed the main issues point by point in our response and provided the necessary supplementary clarifications. If you still have questions or require further information, please let us know before the deadline and we will follow up. If the clarifications above have resolved your primary concerns, we kindly ask that you consider updating your review and/or score. Thank you very much for your time and help.

---

### Official Review · Reviewer_AqD9 · 2025-11-04

**Soundness:** 3
**Presentation:** 3
**Contribution:** 3
**Rating:** 8
**Confidence:** 3

**Summary:**

This paper introduces PixNerd, a end-to-end pixel-space diffusion model that matches the performance of SOTA latent diffusion model without relying on pretrained VAE. PixNerd achieves this by using implicit neural field to replace the traditional decoding head of DiT. PixNerd achieves competitive performance on ImageNet and text-to-image generation tasks.

**Strengths:**

- The paper is well-written with clear structure.
- The paper present comprehensive comparisons against current SOTA models both qualitatively and quantitatively. PixNerd matches or exceeds the performance of comparable methods on ImageNet and text-to-image tasks
- The ablation studies are conducted systematically to evaluate each design choice.

**Weaknesses:**

- The training memory usage is almost doubled compared to that of latent diffusion counter part.
- PixNerd's performance at higher resolutions (512×512) does not scale as strongly as at 256×256. For example, PixNerd is better than SiT-XL on ImageNet256 but falls behind on ImageNet512. Does this imply that the gains from the neural field head diminish at higher resolutions?
- Minors:
	- The citation of Rectified flow seems missing.
	- Table 4 should specify that the comparison is reported on ImageNet256 for clarity.

**Questions:**

- What is the main source of additional training memory and how can this be optimized?
- Why is DCT-Basis encoding better? What about the other popular alternatives like RoPE?
- I notice that PixNerd (512x512) is finetuned from PixNerd(256x256) and its performance on ImageNet512 is not as impressive as PixNerd(256x256). What's the reason for not training PixNerd (512x512) from scratch?
- Is PixNerd compatible with representation alignment techniques like REPA? Since it operates directly in pixel space, would this alignment even be more effective?

---

> ### Author Response · Authors · 2025-11-17
>
> **Q.1 How can additional training memor be optimized?**
>
> We use *torch.compile* to optimize the computation. Traditional PyTorch programs use eager execution, which runs line by line. In contrast, torch.compile traces the model’s forward computation graph, optimizes the graph, and eliminates redundant computations by fusing kernels—for example, AdaLN functions consists of element-wise mul then add, this can be fused into a unified kernel with less memory access. We found that our training process benefits significantly from torch.compile.
>
> **Q.2 Why DCT-Basis encoding?**
>
> DCT is more sensitive for positions than SinCos. We provide the visualization of them at: https://anonymous.4open.science/r/PixNerd_Rebuttal-1EF8/dct_pe.pdf
>
> **Q.3 Why not train PixNerd from scratch on 512 resolution?**
>
> Curriculum learning on resolution is a standard training procedure for diffusion generative models[1, 2, 3], which significantly reduces training resources. We have also demonstrated the feasibility of this approach for Pixnerd in text-to-image (t2i) tasks.
>
> **Q.4 The inferior performance of PixNerd on 512 resolution.**
>
> We attribute inferior performance on 512 resolution to the limited training steps, since we only finetune PixNerd on 512 resolution for 100k steps. We will resume the training from 100K steps to 200K steps. We also adjust the CFG value of the sampling. We achieve better performance than DiT and SiT.
>
> |  |  | **ImageNet 512×512** |  |  |  |  |
> |------------------------------|---------|----------------|----------------|----------------|----------------|----------------|
> | Model | Params | FID↓ | sFID↓ | IS↑ | Pre.↑ | Rec.↑ |
> | **Latent Diffusion Models** |  |  |  |  |  |  |
> | DiT-XL/2 | 675M + 86M | 3.04 | 5.02 | 240.82 | 0.84 | 0.54 |
> | SiT-XL/2 | 675M + 86M | 2.62 | 4.18 | 252.21 | 0.84 | 0.57 |
> | FlowDCN-XL/2 | 608M + 86M | 2.44 | 4.53 | 252.8 | 0.84 | 0.54 |
> | **Pixel Diffusion Models** |  |  |  |  |  |  |
> | ADM-G  | 559M | 7.72 | 6.57 | 172.71 | 0.87 | 0.42 |
> | ADM-G, ADM-U | 559M | 3.85 | 5.86 | 221.72 | 0.84 | 0.53 |
> | RIN | 320M | 3.95 | - | 210 | - | - |
> | SimpleDiffusion | 2B | 3.54 | - | 205 | - | - |
> | **PixNerd-XL/16 (FT100K+CFG 3.5)** | 700M | 2.84 | 5.95 | 245.62 | 0.80 | 0.59 |
> | **PixNerd-XL/16 (FT100K+CFG 4.5)** | 700M | 2.56 | 5.50 | 277.80 | 0.80 | 0.60 |
> | **PixNerd-XL/16 (FT200K+CFG 4.5)** | 700M | 2.45 | 5.19 | 275.77 | 0.79 | 0.63 |
>
> **Q.5 Is PixNerd compatible with representation alignment techniques like REPA?**
>
> Yes, PixNerd is training with REPA as a default configuration. We also provide ablation w/o REPA here.
>
> |          | FID   | SFID  | IS    | Precision | Recall  |
> | -------- | ----- | ----- | ----- | --------- | ------- |
> | w/o REPA | 67.68 | 9.96  | 19.41 | 0.338     | 0.551   |
> | REPA loss weight 0.1 | 42.49 | 10.19 | 37.11 | 0.448     | 0.61    |
> | REPA  loss weight 0.5 **(Default)**   | 37.31 | 10.44 | 42.92 | 0.426     | 0.6275  |
>
>
> **W.1 Missing Reference and Ambiguity**
>
> To enhance clarity, we will carefully revise our paper to specify that the comparison is reported on ImageNet256 and include the reference to Rectified Flow.
>
> [1] Wu, Chenfei, et al. "Qwen-image technical report." arXiv preprint arXiv:2508.02324 (2025).
>
> [2] Esser, Patrick, et al. "Scaling rectified flow transformers for high-resolution image synthesis." Forty-first international conference on machine learning. 2024.
>
> [3] Podell, Dustin, et al. "Sdxl: Improving latent diffusion models for high-resolution image synthesis." arXiv preprint arXiv:2307.01952 (2023).
>
> [4] Liu, Xingchao, Chengyue Gong, and Qiang Liu. "Flow straight and fast: Learning to generate and transfer data with rectified flow." arXiv preprint arXiv:2209.03003 (2022).

---

### Meta-Review · Area_Chair_pBZF · 2025-12-09

**Summary:**

PixNerd proposes a single-stage, end-to-end pixel-space diffusion transformer  without using VAEs or multi-stage cascades. The key idea is to keep computation cheap with large patches, but replace the usual linear projection head with a patch-wise neural-field decoder for details.

**Reviewer Concerns:**

In the revision, the authors directly address two reviewer concerns with new experiments: (1) they re-evaluate and mitigate the previously noted weaker 512×512 performance (AqD9, vPnv), and (2) they add distillation results for PixNerd (iKuE). Together, these additions strengthen the empirical case for the method.

**Reviewer Scores:**

Most reviewers are likely to keep their scores in the accept-leaning range.

---

### Decision · Program_Chairs · 2026-01-26

Accept (Poster)